# Two modes of evolution shape bacterial strain diversity in the mammalian gut for thousands of generations

N. Frazão [1], A. Konrad[1,3], M. Amicone [1], E. Seixas [1], D. Güleresi [1], M. Lässig [2] & I. Gordo [1]

How and at what pace bacteria evolve when colonizing healthy hosts remains unclear. Here, by monitoring evolution for more than six thousand generations in the mouse gut, we show that the successful colonization of an invader *Escherichia coli* depends on the diversity of the existing microbiota and the presence of a closely related strain. Following colonization, two modes of evolution were observed: one in which diversifying selection leads to long-term coexistence of ecotypes and a second in which directional selection propels selective sweeps. These modes can be quantitatively distinguished by the statistics of mutation trajectories. In our experiments, diversifying selection was marked by the emergence of metabolic mutations, and directional selection by acquisition of prophages, which bring their own benefits and costs. In both modes, we observed parallel evolution, with mutation accumulation rates comparable to those typically observed in vitro on similar time scales. Our results show how rapid ecotype formation and phage domestication can be in the mammalian gut.

The well-known capacity of bacteria to adapt to laboratory environments[1–3], as well as the high level of evolutionary parallelism that is observed when bacteria are exposed to the same stresses ex vivo, demands an understanding of their evolutionary change in natural ecosystems. Time series metagenomics data have started to reveal that significant amounts of evolution can occur in the microbiomes of every human throughout their lifetimes[4,5]. These genetic changes lead to individualized microbiomes and may provide insight into the main drivers of colonization success for an invading strain or species, with consequences for pathogenesis[6]. Microbiome evolution studies within hosts may provide answers to questions such as: What are the selection pressures operating in the gut? How is bacterial evolution influenced by complex communities[7,8]? How do the different evolutionary mechanisms – mutation, genetic drift, recombination, and natural selection – shape the diversity of the gut ecosystem[4]?

Here we leverage on an experimental system previously used to study short-term evolution in a minimally perturbed microbiome[9] to enquire about rates and fitness effects of mutations in *Escherichia coli* during long-term evolution in the mouse gut. Mice can be used as models for health problems relevant to humans[10] and help to understand how microbial traits may cause host phenotypes[11]. While selection for metabolic mutations and events of phage-driven horizontal gene transfer were shown to occur during the first weeks of *E. coli* colonization[9,12,13], the consequences of these mechanisms for its tempo and mode of evolution along the host life are still unknown. Here we show that *E. coli* follows a clock-like rate of adaptive molecular evolution, prophage induction evolves to optimal levels and different *E. coli* lineages converge evolutionarily in the mammalian intestine. We uncover that natural selection within a host can act in two modes: diversifying or directional selection, each entailing different fitness consequences when evolved clones transmit to new hosts.

[1]Instituto Gulbenkian de Ciência, Oeiras, Portugal. [2]Institute for Biological Physics, University of Cologne, 50923 Cologne, Germany. [3]Present address: CE3C – Center for Ecology, Evolution and Environmental Changes, Faculdade de Ciências da Universidade de Lisboa. Campo Grande, Lisboa, Portugal. ✉e-mail: nfrazao@igc.gulbenkian.pt; igordo@igc.gulbenkian.pt

## Results

### Rate of mutation accumulation in invading *Escherichia coli*

We report an in vivo long-term experiment that maps the evolution of an invading *Escherichia coli* strain in the presence of a diverse mouse microbiota over thousands of generations. The invasion becomes possible after a short perturbation by an antibiotic[9] (Fig. 1 and Supplementary Fig. 1, Supplementary Data 1–3). We colonized nine mice with an invading antibiotic-resistant *E. coli* carrying a fluorescent marker to track its long-term evolution while resident in the mouse gut (see Methods). Five out of the 9 mice were already followed, in the short-term (27 days), in a previous published study, where we found that these prophages carry metabolic benefits[9]. In two mice, colonization with the invading *E. coli* was unsuccessful. We found that the Shannon index is negatively correlated with the invader long-term *E. coli* abundance (Log10(CFUs)) (r = −0.78 ((−0.85, −0.67), 95%CI), df = 77, p < 0.0001) and persistence (1-presence or 0-absence) (−5.892 (±1.902), p = 0.002), but not with the presence of a resident *E. coli* belonging to a different phylogenetic group (B1)[9] (Fig. 1a, Supplementary Fig. 2, Supplementary Data 2). Interestingly, we found that the resident *E. coli* persists independently of microbiota diversity. This highlights the need to understand diversity at the strain level to predict

key functions of the microbiome, such as its capacity to provide colonization resistance to opportunists[14].

We followed the long-term evolution of invader *E. coli* (Fig. 1a, Supplementary Data 1) by pool-sequencing of clones sampled from each mouse's feces. In three mice, long-term colonization (>6000 generations) was achieved, while shorter-term colonization occurred in the other mice. We estimated 15 generations/day for the invader *E. coli* by measuring bacterial ribosomal content (Supplementary Fig. 3 and Supplementary Data 4). Temporal series sequencing allowed estimation of the rate of molecular evolution from the accumulation of novel mutations in the invader *E. coli* genome, isolated from each mouse. Across mice, *E. coli* evolution was characterized by an elevated ratio of non-synonymous to synonymous mutations, indicative of adaptive evolution driven by strong selection (Fig. 1b, Supplementary Data 5). The overall mutation accumulation rate is remarkably similar to Lenski's long-term evolution of non-mutator *E. coli* in vitro[1], despite differences in ecology (Fig. 1c and Supplementary Data 6). Across all mice, mutations accumulated at an average rate of $2.1×10^{-3}$ ($0.53×10^{-3}$ 2SE) per genome per generation, suggestive of a clock-like rate of adaptive evolution within a healthy host.

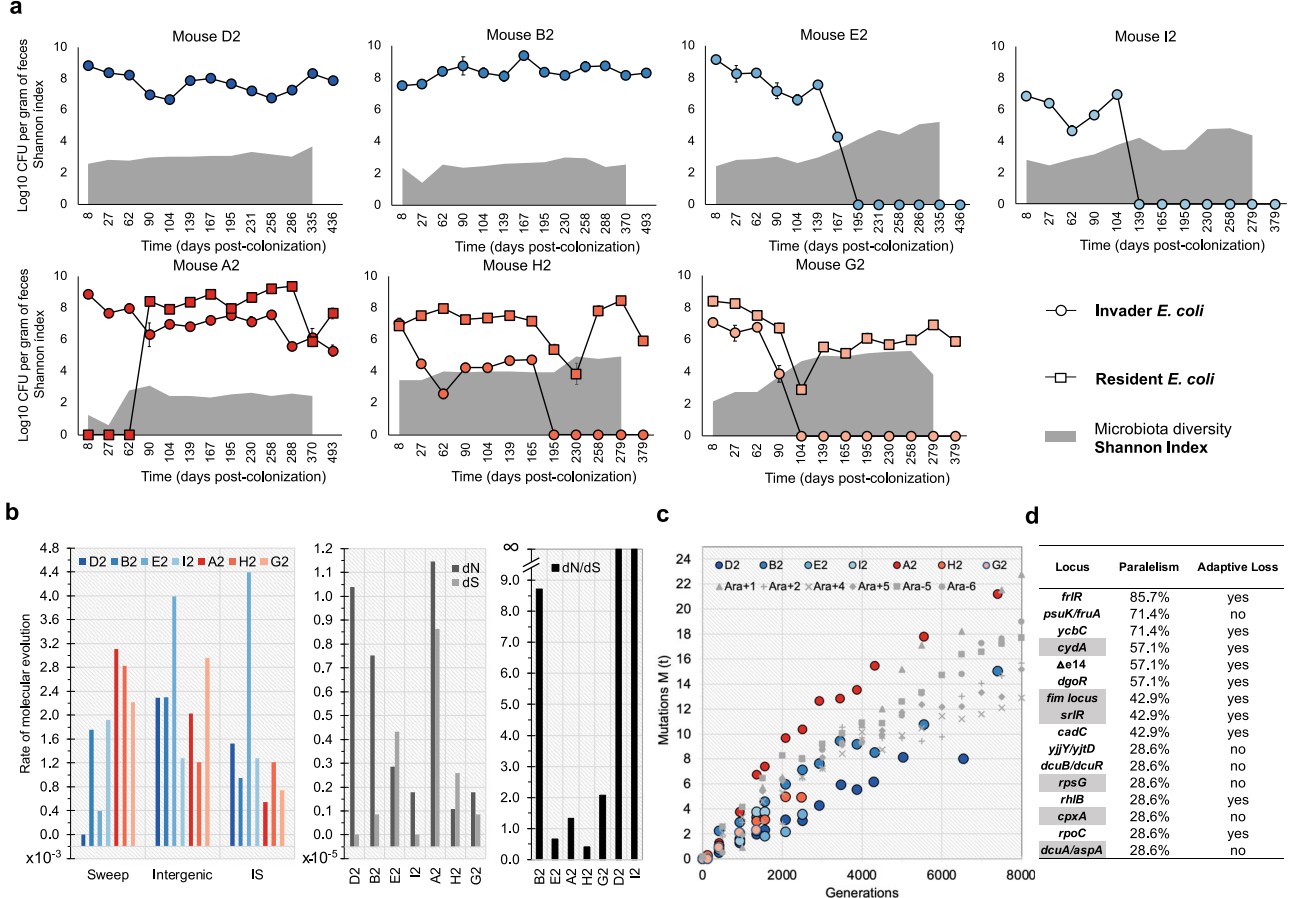

**Fig. 1 | Colonization success depends on microbiota diversity and rates of molecular evolution across mice. a** Time series of the abundances of the invader (circles) and resident (squares) *E. coli* lineages. Each circle represents the mean value and error bars represent 2*Standard Error (2SE) (*n* = 3 technical replicates, Supplementary Data 1). Time series of the microbiota species diversity (Shannon diversity). The values of microbiota diversity observed are shown as a gray area. **b** Rate of molecular evolution, per generation, in each host (sweeps, intergenic mutations, Insertion Sequences). dN and dS, and dN/dS ratio. **c** *E. coli* mutation accumulation during in vivo and in vitro evolution (data of non-mutator lines

from[1]). "Mutations M(t)" correspond to the sum of allele frequencies at each sampling point for the in vitro and in vivo experiments (Supplementary Data 6 and 8 to 14). Blue and red hues represent mice where the invader colonized the gut alone or together with the resident *E. coli* lineage, respectively. **d** Level of mutational parallelism in the invader lineage across hosts: the genetic targets of adaptation and the frequency of hosts where it was changed is shown, as well as the changes that likely involve loss of function. Mutation targets also found in Lenski's in vitro evolution experiment, data from[1], are highlighted in gray.

We also detected signs of parallel evolution *i.e.*, changes in the same mutational targets[15] within and across mice. The level of parallel evolution varied across gene targets: the locus with the highest parallelism (mutated in *E. coli* populations isolated from six out of seven mice) was *frlR*, which was also found as an adaptive mutation in the previous studies[9,16]. In contrast, a target of mutation, also observed previously[17], the locus *dgoR*, was mutated only when the resident *E. coli* was absent (Fig. 1d, Supplementary Data 7–14). This may be explained by the different metabolic capacities between the resident and invader: the resident lacks the genes to consume fructoselysine but both can consume galactonate (Supplementary Data 15). Evolutionary parallelism was also observed at the insertion sequence (IS) element level, indicating that many of these changes are adaptive (see below and Supplementary Data 8–14). Across all mice, 277 evolutionary changes were detected, some of which were beneficial (Fig. 1d), while others could be neutral or even slightly deleterious mutations that 'hitchhiked' with adaptive ones (Supplementary Data 5 and 8 to 14). Of these, 24% (66/277) occurred in intergenic regions, indicating that alterations in gene regulation occurred during invader *E. coli* adaptation to the gut. Changes in intergenic regions, such as *yjjY/yjtD* and *psuK/fruA*, can alter the regulation of flanking genes by altering gene transcription and translation (e.g., *arcA*, *yjtD* and *psuK* gene expression), which may drive evolutionary adaptation (Supplementary Data 16). Over the cumulative >29,000 generations of evolution, 46 selective sweeps were observed. We defined a selective sweep as a mutational trajectory that reached >95% frequency and remained at these high frequencies for the duration of observation (Supplementary Data 5). These sweeps were caused by de novo substitutions, indels, ISs, and horizontal gene transfer (HGT) events, mediated by phages and plasmids (Supplementary Data 8–14). Together, our observations show that an invader *E. coli* followed fast, adaptive evolutionary dynamics in the gut environment for thousands of generations.

## Diversifying and directional selection

The large-scale dynamics of molecular evolution and its effects on the diversity of the invading strain populations, however, play out in strikingly different ways across the individual mice. Specifically, we observed two modes of evolution: one in which new ecotypes form in the invader *E. coli* and are maintained for >6000 generations (Fig. 2), and a second, in which recurrent selective sweeps intertwined with HGT events (Fig. 3).

Mouse D2 developed an ecotype characterized by an unprecedented long-term coexistence of invader sublineages within a species-rich gut microbiome. No single mutation fixed over ~6540 generations, even though the total population size of the invader *E. coli* was large (~$10^7$–$10^8$ CFUs/g). The lack of fixations indicates that diversifying selection dominates over the directional selection and maintains genetic variation over extended periods. In mice E2 and B2, we also observe signals of diversifying selection and prolonged polymorphism lifetimes. However, the coexistence of ecotypes is only transient and overridden by occasional fixations (in mouse B2 after 6000 generations, in mouse I2 after 1500 generations), in accordance with model results[18].

The first mutation to arise in the D2 invader *E. coli* was a single-nucleotide polymorphism (SNP) (Supplementary Data 10), lying in the promotor region of *psuK*, which encodes a pseudouridine kinase. This SNP alters *E. coli* growth in vitro (Supplementary Fig. 4, Supplementary Data 17). Despite reaching high frequency (85% by day 62), it failed to reach fixation over more than a full year of evolution in the D2 mouse. Mutations in *dgoR* may have been the mechanism causing diversifying selection in mouse D2 and reducing the sweep rates in other mice (Fig. 2, Supplementary Data 9, 11 and 14). Specifically and consistent with previous results[13], we found signals of negative-frequency-dependent-selection acting on the *dgoR* mutations in mouse D2 and E2 (Supplementary Fig. 5a, b, Supplementary Data 7), where *dgoR* KO is

advantageous when rare but deleterious at high frequency. Furthermore, competitive fitness assays in mice showed that the fitness effect of a *dgoR* KO depends on the presence of the resident, as expected by the resident's metabolic capacity to compete for galactonate and thus drive the concentration of this resource down (Supplementary Fig. 6, Supplementary Data 15). In one-to-one competition experiments in the mouse gut, the *dgoR* KO mutant decreases to very low frequency when the resident is present but only to ~10% when it is absent (linear mixed model followed by anova test: *dgoR* KO mutant frequency in presence versus absence of resident, $P < 0.0001$; Supplementary Fig. 7, Supplementary Data 18). Evidence of strong selection to consume galactonate is further provided by the observation of recurrent mutations in *dgoR* (Supplementary Data 10), both in our mice and in other work on *E. coli* strains adapting to the gut of streptomycin-treated mice, where a resident is absent[13,17]. The combined evidence that disruption of *dgoR* is under negative-frequency-dependent-selection, whose strength changes in the presence of the resident, provides a plausible explanation for the long-term maintenance of *E. coli* polymorphism in mouse D2 even in the face of other mutations[18] and for the absence of *dgoR* mutations in the mice with the resident. Negative-frequency-dependent-selection was also observed in mouse B2 for *srlR*. Here, mutations altering competition for sorbitol (Supplementary Data 9) also have an advantage when rare (~0.4% per generation) but are a disadvantage when common (Supplementary Fig. 5c), consistent with previous findings in streptomycin-treated mice[19].

Mouse A2, H2, and G2 exhibited evolution under directional selection (Fig. 3). A priori, the presence of a closely related strain in these mice could alter the mode of invader evolution in different ways: by reducing population size and slowing down adaptation, by reducing the number of available nutritional niches[20] (Supplementary Data 15), or by increasing selection strength and enhancing the rate of fixations. In populations under directional selection and a low mutational input or a higher selective pressure, periodic sweeps should occur (mouse H2). Contrarily, in populations where the mutational input is higher, or the selective pressure is lower should lead to numerous beneficial mutations competing for fixation, which originate clonal interference (mice A2 and G2). In our experiments, the sweep rate was not correlated with the mean population size, but was significantly higher when co-existence between invader and resident occurred (Welch two-sample t-test t = 2.88, df = 4.23, $P$ value = 0.021, one sided). This indicates that the sweep pattern is driven by selection induced by the resident, which we previously showed to carry prophages that can attack the invader[9]. Phage attack and transfer could provide a mechanism by which new selection pressures would alter the mode of long-term evolution for the invader *E. coli*. Indeed, co-existence of the resident and invader *E. coli* strains resulted in the co-occurrence of selective sweeps and two distinct mechanisms of HGT by transduction and conjugation (Fig. 3a, Supplementary Fig. 8, Supplementary Data 5 and 8). During the first thousand generations, when the invader lineage was dominant in mouse A2, seven mutations accumulated and reached high frequency. After this period, and with the concurrent rise in frequency of the resident *E. coli* (Fig. 1a), a strong interlude of HGT marked the evolution of invader *E. coli* in the gut. This occurred initially by phage-mediated transfers, with the rapid formation of a double lysogen, followed by the acquisition of two plasmids from the resident strain (a large plasmid of ~109Kb and a smaller plasmid of ~69Kb) (Fig. 3a). The process of phage-driven HGT was observed in all the mice where invader and resident *E. coli* co-existed (Supplementary Data 8, 12 and 13, Fig. 3). Interestingly, of the seven putatively active prophages in the resident, only two (Nef and KingRac) transferred to the invader *E. coli* genome during long term co-existence, and the double lysogens that formed rapidly reached high frequencies and were maintained in the gut for a long time.

Conjugation was only detected in mouse A2 and only the small plasmid was maintained for over 7000 generations, being detected in

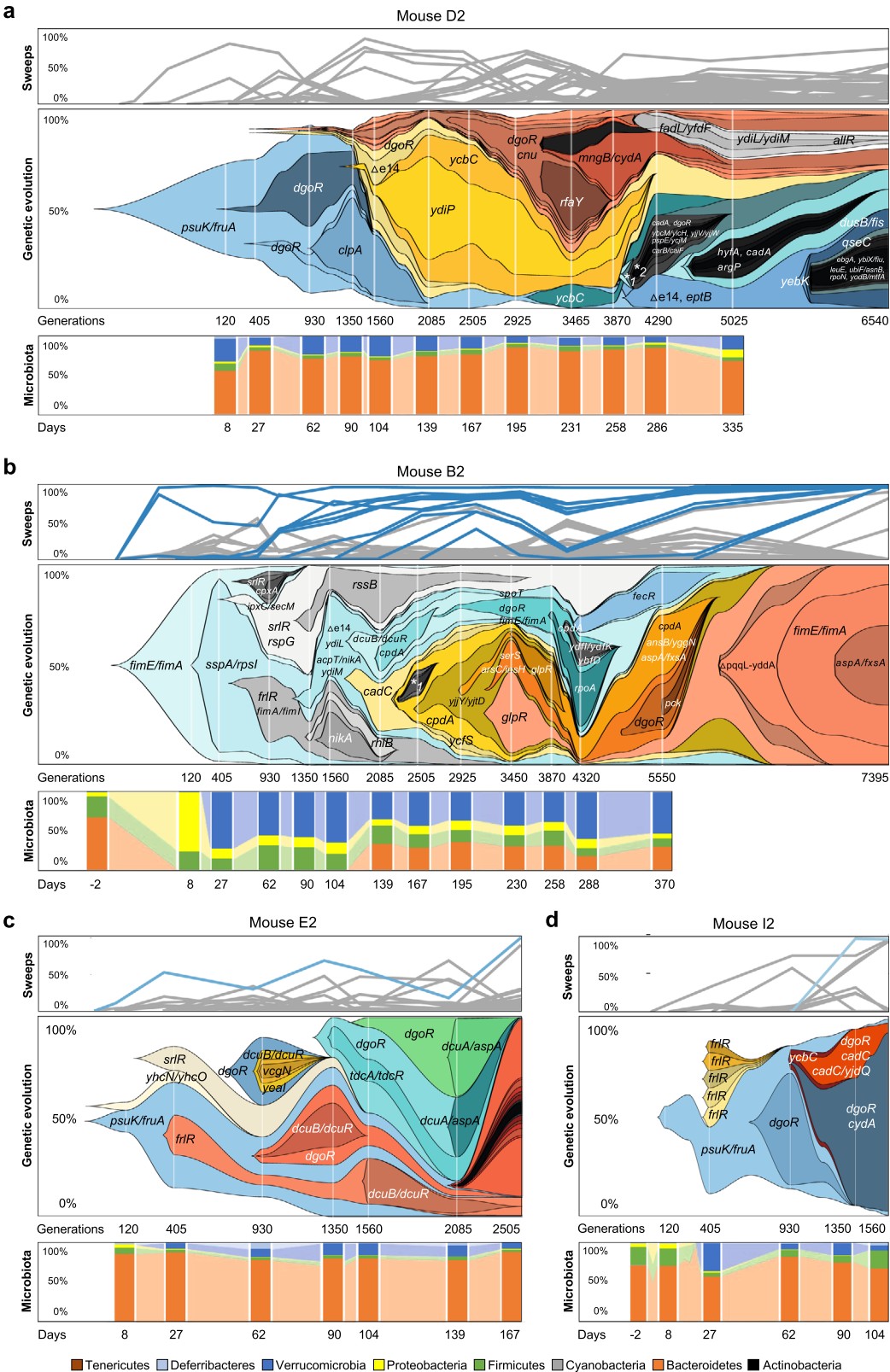

**Fig. 2 | Evolution of invader *E. coli* in the absence of the resident. a−d** Frequency of the mutations identified in the invader *E. coli* population across time in mouse D2, B2, E2 and I2 (top panels) and corresponding Muller Plots showing the spread of mutations observed in each mouse (middle panels). Microbiota composition (phylum level) along time (bottom panels), showing a high temporal stability in the long term. *1 *fimA, frlR, ykfI, dgoR, dcuA/aspA, qseC, cydA;* *2 *ycbC, rpoC, frlR.* Mutations that reached frequency above 95% are highlighted in color, other frequency trajectories shown in gray (Supplementary Data 5, 9, 10, 11 and 14).

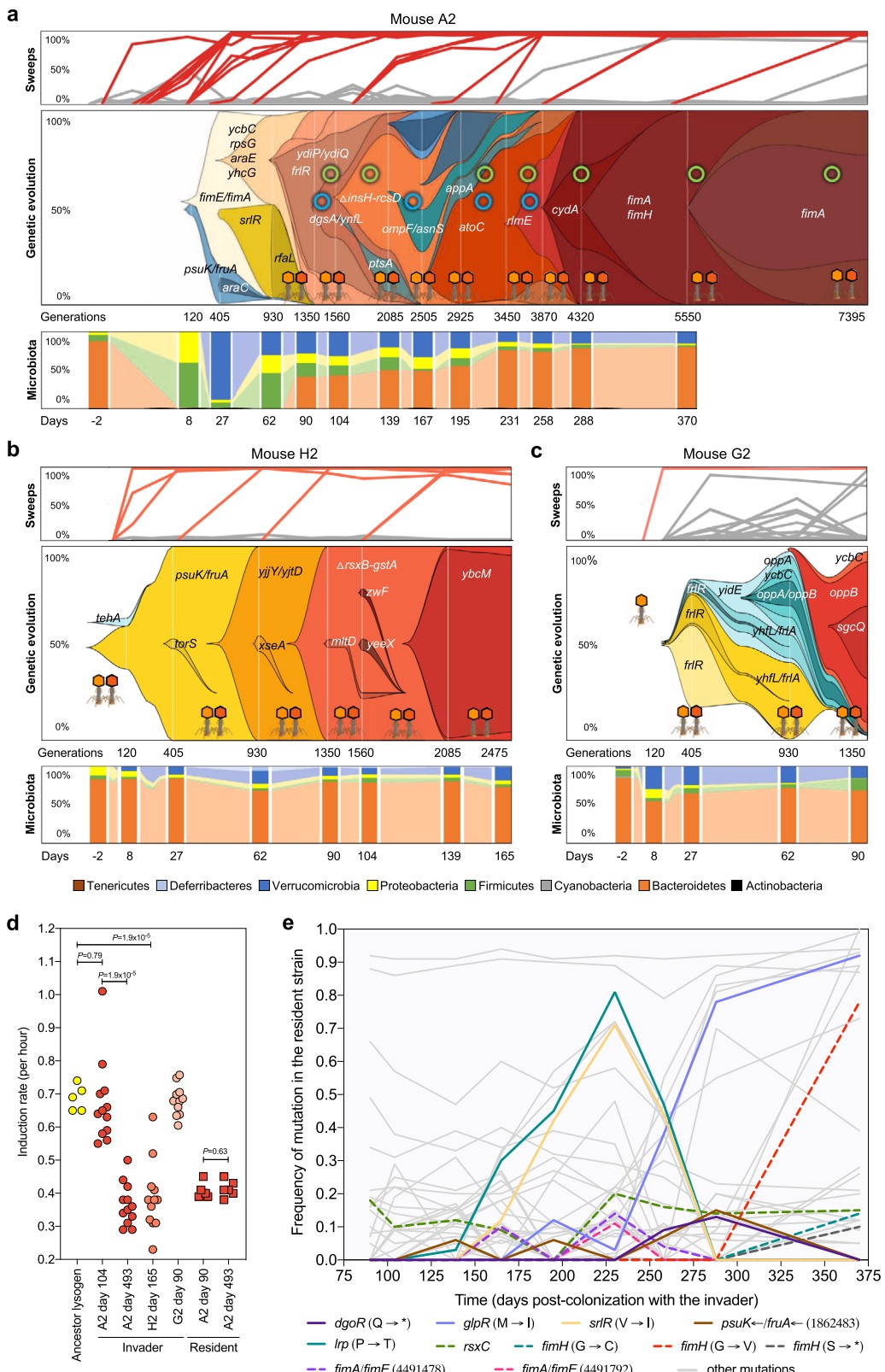

96% of the clones at day 493 (Supplementary Fig. 8 and Supplementary Data 19). This indicated that any potential benefits that the large plasmid could bring[21] were outweighed by its cost in the invader *E. coli* background. The context dependency of plasmid fitness has also recently been seen across gut isolates[21]. Overall, 22 selective sweeps accounted for the evolution of invader *E. coli* in mouse A2 for >7000

generations; in mouse H2, seven sweeps were observed in <3000 generations (Fig. 3 and Supplementary Data 5, 8 and 13).

Phage-driven HGT was a hallmark of the mode of evolution with the resident *E. coli* (Fig. 3). These phages, when fixed, bring metabolic benefits to the invader[9], but induction into a lytic cycle can also entail fitness costs as it leads to the lysogens' death. In vitro, the induction

**Fig. 3 | Evolution of invader *E. coli* when co-existing with the resident strain.**
**a**–**c** Frequency of the mutations identified in the invader *E. coli* population across time in mouse A2, H2 and G2 (top panels), the corresponding Muller Plots (middle panels) and microbiota compositions at the phylum level (bottom panels). The spread of de novo mutations is indicated by the gene targets where they emerged, intertwined with HGT events, mediated by two phages (in light and dark orange) and two plasmids (green circle – small plasmid: ~69Kb; blue circle – big plasmid: ~109Kb) acquired from the resident strain which started to be detected on day 90 in mouse A2 but much earlier in the other mice. The frequency of phage- or plasmid-driven HGT events can be found in Supplementary Data 8, 12, 13, and 19. In mouse H2 invader evolution could be followed along 2475 generations, after which extinction occurred and in mouse G2 evolution could be followed for 1350 generations. Mutations that reached frequency above 95% are highlighted in color, other frequency trajectories are shown in gray (Supplementary Data 5, 8, 12 and 13). **d** Evidence for prophage domestication, shown by a reduction of lytic induction during gut adaptation. The ancestral lysogen and clones randomly sampled from mouse A2 at days 104 and 493, and mouse H2 at day 165, which bear Nef and KingRac prophages (lysogens), had their prophage induction rate (per hour) measured in a mitomycin C assay. *P* values were calculated by a two-sided t-test (*n* = 4 biologically independent replicates per clone tested, Supplementary Data 18). **e**, Molecular evolution of the resident strain in mouse A2. The loci highlighted in color were targets of evolution in both the resident and the invader *E. coli* strains. Mutations hitting genes with metabolic functions are indicated with solid lines (*i.e. dgoR, glpR, srlR, psuK/fruA* and *lrp*).

rate for lysogens of the invader *E. coli* was higher than that of the resident strain (Fig. 3d) and in vivo, induction rates are likely to be even higher, as it is easier to observe phage-driven HGT in the mice than in the Petri-dish[22]. Theoretical modeling of lysogens with different induction rates predicts that selection should lead to a reduction of the induction costs experienced by the invader *E. coli* during its long term evolution with the resident *E. coli* strain (if the resident has a lower induction rate)[9,23]. Clones isolated just after the acquisition of the two prophages (at day 104, mouse A2) (Fig. 3a) have a mean induction rate of 0.67 h$^{-1}$ (0.07 2SE), similar to that of a newly formed double lysogen (0.69 h$^{-1}$ (0.02 2SE) (Fig. 3d, Supplementary Data 20). Clones sampled after long-term evolution (at day 493 post-colonization in mouse A2, or day 165 in mouse H2), showed a significantly lower rate of lysogen induction: 0.37 h$^{-1}$ (0.04 2SE) for mouse A2 and 0.39 (0.06 2SE) for mouse H2, (Unpaired t test: $P = 2\times10^{-7}$ and $P = 9\times10^{-5}$, respectively) (Fig. 3d, Supplementary Data 20). Furthermore, the induction rate of the resident *E. coli* clones was maintained through time (0.41 h$^{-1}$ (0.01 2SE)) (Fig. 3d, Supplementary Data 20). These results indicate that under evolution with the resident *E. coli*, lysogens in the invader population adapt over a few thousand generations to avoid the high costs of induction occurring after they are formed and evolve an optimal level of induction. Given that no mutations were detected in the prophage sequences, the observed reduction of lytic induction was likely caused by sweeps that accumulated in the bacterial chromosome. This indicates epistasis between prophages and mutations, as some of the accumulated mutations may be beneficial only in the presence of the prophages. Thus, *E. coli* can 'domesticate' its phages to maintain the benefits they bring and avoid the costs of induction into a lytic cycle. We hypothesise that, in the long-term, epistasis between prophages and background mutations to fine-tune prophage induction to optimal levels could be one of the drivers of directional selection in the gut. This may help explain why it is difficult to induce phages from gut isolates of *E. coli* strains we used in this experiment.

## Evolutionary convergence between *E. coli* strains

Many of the adaptive mutations observed (e.g., frameshifts and missense mutations in *frlR, srlR, dgoR, kdgR*) likely caused changes in the ability of the invader to consume different resources, supporting the notion that adaptation to different resources is important for structuring the microbial diversity in the intestine[24,25]. Given that resource competition can also shape the evolution of the resident *E. coli*[26], we pool-sequenced its clones from mouse A2. Aligning the reads against the reference genome revealed 47 mutations in the resident *E. coli* (Fig. 3e, Supplementary Data 21). Of the mutations observed, 28% occurred in genetic targets that were also seen in the invader *E. coli* strain in mouse A2 and others (Fig. 3e, Supplementary Data 8 to 14, 21 and 22). Possibly, the *E. coli* strains share similar niches and influence each other's ability to form stable ecotypes. An example of this effect is detectable in the time series obtained for the two *E. coli* strains in mouse A2: in the invader lineage, the frequency of the *srlR* mutation rose between days 27 and 62, then decreased to undetectable levels by day 90 (Fig. 3a, Supplementary Data 8). Likewise, a mutation in *srlR* was

detected in the resident *E. coli* by day 165, reached 71% frequency by day 230, but by day 288 was no longer detected (Fig. 3e, Supplementary Data 21). Similarly, the intergenic region *psuK/fruA* was a target of selection across invader *E. coli* populations isolated from different mice (71%, Supplementary Data 7) and a target in the genome resident *E. coli* of mouse A2 (Fig. 3e). The *glpR* gene was also mutated in the resident of mouse A2 (Fig. 3e, Supplementary Data 21 and 22) and was a target of selection in invader *E. coli* in another animal (Supplementary Data 9).

Beyond metabolic evolution, mutations in targets related to cell structure were also convergent between invader and resident bacteria. The *fim* operon was recurrently mutated throughout the colonization in both invader and resident strains (Fig. 2b, and Fig. 3a, e). Mutations occurred first in the *fimE/fimA* intergenic region, then in the structural genes *fimA* and/or *fimH* of the invader or the resident. The *fimH* gene is highly polymorphic in *E. coli*[27] and an important contributor to *E. coli* pathogenesis[28]. Changes in the *fim* locus can lead to changes in motility or biofilm formation[28]. In vitro assays showed that an invader-evolved clone, which we sequenced (Supplementary Data 23), presented reduced motility (Supplementary Fig. 9a), but not biofilm capacity (Supplementary Fig. 9b, Supplementary Data 24), and when growing under static conditions formed aggregates of cells at the liquid-air interface (Supplementary Fig. 10a, b). This evolved phenotype is similar to that observed during in vitro adaptation of *E. coli* to a fluctuating environment[2] and indicates that the A2 invader lineage may have evolved an aggregation trait to occupy a different niche than its ancestor or the resident, where the trait did not evolve (Supplementary Fig. 10c, d). The changes at the *fim* locus and other loci important for *E. coli* metabolism (Supplementary Data 8 and 21) indicate evolutionary parallelism occurred in both invader and resident strains. These data illustrate the potential for a high level of evolutionary convergence between two phylogenetically distinct strains colonizing the intestine of the same host.

## Frequency-time statistics of mutational trajectories

To quantitatively characterize the modes of evolution and underlying selection, we used the time-resolved mutation trajectories resolved in Figs. 2 and 3 to estimate the probability that a mutation reaches frequency $x$ and the average time it takes to reach frequency $x$. These functions, $G(x)$ and $T(x)$, are discussed in Methods and plotted in Fig. 4a, b and Supplementary Fig. 11. Next, we evaluated two simple summary statistics: the probability that a mutation established at an intermediate frequency reaches near-fixation, $p = G(0.95)/G(0.3)$, and the ratio of average times to near-fixation and to an intermediate frequency, $\tau = T(0.95)/T(0.3)$. The threshold frequencies used here are chosen for practical purposes; their details do not matter for the subsequent results. The joint statistics of fixation probabilities and times allow three scenarios of adaptive evolution to be distinguished: diversifying selection leading to (i.) coexistence of ecotypes and directional selection leading to (ii.) clonal interference or (iii.) periodic sweeps (Fig. 4c and Methods). Under predominantly diversifying selection, few established mutations fix and relative fixation times are

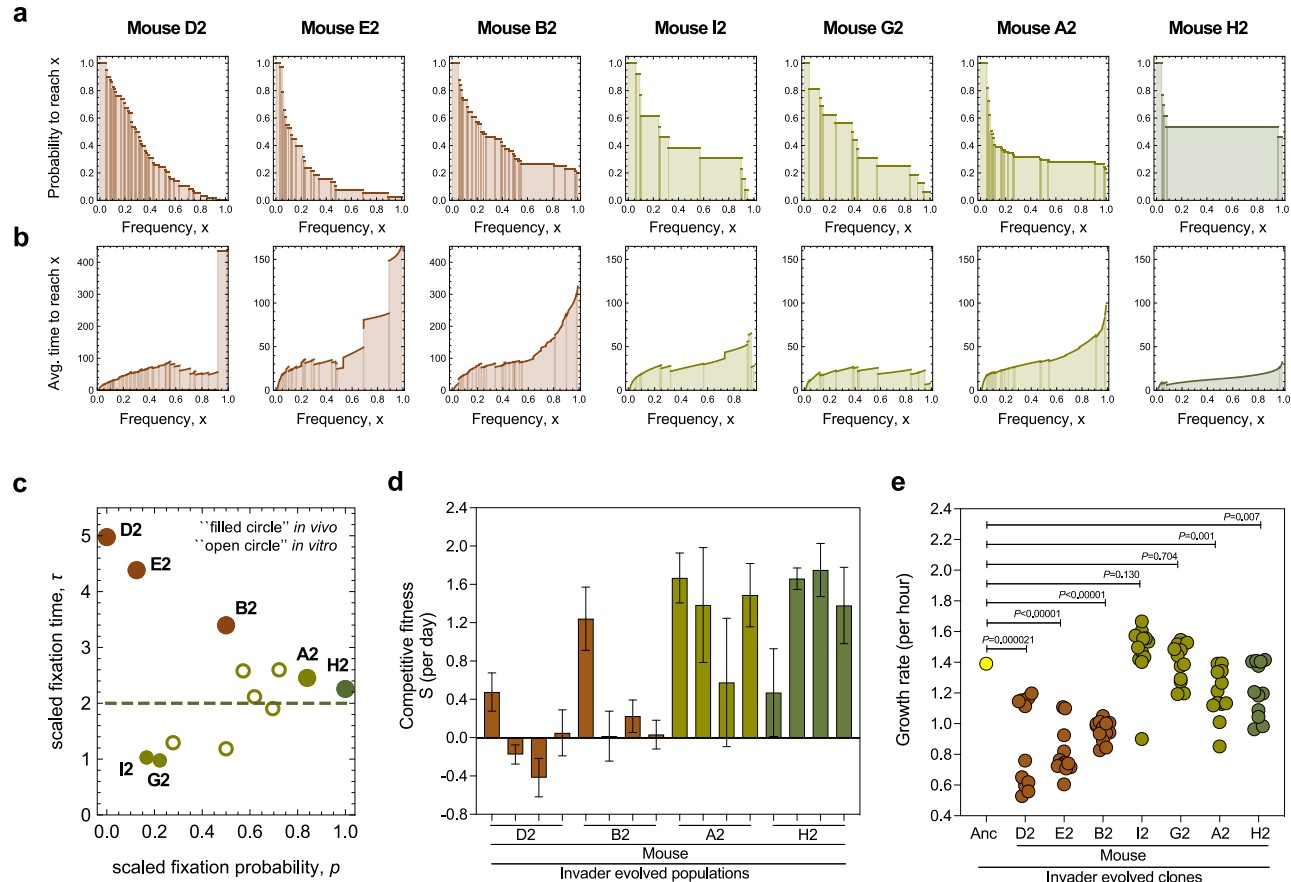

**Fig. 4 | Modes of evolution and fitness tradeoffs. a–c** A $p$-$\tau$ selection test captures the joint statistics of fixation probabilities **a**, and times **b**. This test **c**, identifies dominant diversifying selection in mice D2, B2, and E2 (brown), and directional selection generating clonal interference in mice A2, G2, and I2 (green) or periodic sweeps in mouse H2 (dark green). In mouse D2, where no fixations occur, we use the time to the end of the experiment lower bound for $\tau$. The reference value $\tau \approx 2$ expected for directional selection in the low-mutation regime is shown as dashed line. Short-term trajectories (<2000 generations) are marked by small symbols. The data from Lenski's in vitro evolution experiment[1], for the same lines as in Fig. 1, are shown as open circles. **d** Host specificity of in vivo adaptation. Relative fitness of pools of evolved clones from mice D2, B2, A2 and H2 when competing against the ancestor in new mice ($n = 4$, 2 male and 2 female per competitive fitness assay of each of the independently evolved populations). The height of each bar represents the value of competitive fitness of the evolved clones in each mouse, measured as the slope of the linear regression on Ln(CFUevolved/CFUancestor) over the first 4 days after gavage. The error bars represent the standard error of the linear regression slope. **e** In vivo adaptation leads to growth trade-offs in vitro. Growth rate (per hour) measured in LB medium (3–4 replicates) of randomly isolated invader *E. coli* clones evolved in the gut of mice D2, B2, E2, I2, A2, H2 and G2 during 436, 493, 167, 104, 493, 165 and 90 days, respectively. $P$ values were calculated by a two-sided one-sample t-test ($n = 3$ to 4 biologically independent replicates per clone tested, Supplementary Data 24).

long ($p \ll 1, \tau \gg 1$). This mode is consistent with the data from mice B2, and E2. Under clonal interference driven by directional selection, only some fraction of the established mutations fixes and relative fixation times are short ($p < 1, \tau < 2$). This mode was observed in mice A2. In mouse G2 and I2 the duration of the colonization is too short to accurately establish the mode of selection. For periodic sweeps to occur, all established mutations fix and the time to near-fixation is about twice the time to half-fixation, given that individual mutations under positive selection follow a sigmoid frequency trajectory ($p \approx 1, \tau \approx 2$). This mode was found in mouse H2 (Fig. 4c). Together, the $p$-$\tau$ test infers a dominant mode of evolution occurred in mice D2, B2, A2 and H2. These findings are in line with results on *Pseudomonas fluorescens*, where adaptive diversification depends on the level of community diversity[20] and negative-frequency-dependent-selection operates between evolved morphotypes in the absence, but not in the presence of a natural community[29].

To understand if fitness tradeoffs could result from the specific in-vivo modes of evolution, we performed fitness assays in new mice, which will likely have a distinct microbiome composition, and in simpler in vitro environments. First, we performed in vivo competition experiments between evolved populations and the ancestral clone (Fig. 4d, Supplementary Data 25). Only under directional selection, the

evolved populations acquired a net benefit over the ancestral ones. This indicates that when evolution is dominated by diversifying selection the competitive fitness of the populations is more host-specific than when evolution is dominated by directional selection. Second, the populations evolved under diversifying selection show an average 51% reduction of in vitro growth compared to the ancestral K12 strain (Fig. 4e, Supplementary Data 26). K12 is a laboratory-adapted strain first isolated from the mammalian gut[30]. The cost observed here reflects fitness tradeoffs between the laboratory environment and rewilding in the gut. Again, this effect is stronger under predominantly diversifying selection than under directional selection.

The variability of selection regimes of *E. coli* in vivo is higher than in the long-term in vitro experiment conducted over the years by Lenski's team[1]. Here we find a rapid onset of diversifying selection or predominantly directional selection. In contrast, all in-vitro non-mutator populations show a common pattern of clonal interference under strong directional selection (Fig. 4c and Supplementary Fig. 12); at a comparable speed of molecular evolution (Fig. 1c), diversifying selection emerges only at later times[1]. Of course, more in vitro and in vivo experiments are needed to understand the relative likelihood of these evolutionary modes under environments with different number of resources and microbiomes with different levels of complexity.

## Discussion

In summary, our results offer a longitudinal view of *E. coli* long-term evolution within the complex ecosystems of the mammalian gut. Over thousands of generations, metabolic adaptation continuously generates diversifying selection, which can induce the formation of ecotypes and the maintenance of functional and genetic diversity within an invader lineage. The dynamics of diversifying selection can be overridden by unconditionally beneficial mutations or by the presence of a competing conspecific strain, which causes phage-induced directional selection towards prophage domestication. In this case, we observe ecotype competition between invader and resident *E. coli* strains, marked by evolutionary convergence between phylogenetic distinct strains. How the relative frequency of the two modes of evolution identified here shape microbiome diversity and the emergence of mutators affects the selection modes[31] in the gut are important questions for future work.

We conclude that both metabolic interactions and domestication of newly acquired phages shape the mode of evolution of *E. coli* strains inhabiting the mammalian gut. These evolutionary processes, in turn, determine stability, turnover, and diversity of ecotypes.

## Methods

### Ethical statement

This research project was ethically reviewed and approved by the Ethics Committee of the Instituto Gulbenkian de Ciência (license reference: A009.2018), and by the Portuguese National Entity that regulates the use of laboratory animals (DGAV - Direção Geral de Alimentação e Veterinária (license reference: 008958). All experiments conducted on animals followed the Portuguese (Decreto-Lei n° 113/2013) and European (Directive 2010/63/EU) legislations, concerning housing, husbandry and animal welfare.

### Escherichia coli clones

The ancestral invader *E. coli* strain expresses a Yellow Fluorescent Protein (YFP), and carries streptomycin and ampicillin resistance markers for easiness of isolation from the mouse feces [*galK*::amp (pZ12::PL*lacO*−1-YFP, strR (*rpsl*150), Δ*lacIZYA*::scar]. An *E. coli* strain used for the in vivo competition experiments is isogenic to the ancestral invader but expresses a Cyan Fluorescent Protein (CFP) and carries streptomycin and chloramphenicol resistance markers [*galK*::chlor (pZ12::PL*lacO*−1-CFP, strR (rpsl150), Δ*lacIZYA*::scar]. The resident *E. coli* lineage was isolated from the feces along time using McConkey + 0.4% lactose medium, as previously described[9]. All the resident clones sampled from each mouse belong to *E.coli* phylogenetic group B[9]. The invader *E. coli* strains (YFP and CFP) derive from the K-12 MG1655 strain (DM08) and exhibit a gat negative phenotype, *gatZ*::IS1[12]. The resident *E. coli* clone used for the competition experiments in the mouse gut expresses a mCherry fluorescent protein and a chloramphenicol resistance marker, allowing to distinguish the invader and resident strains in the mice feces.

*E. coli* clones were grown at 37 °C under aeration in liquid media Luria broth (LB) from SIGMA − or McConkey and LB agar plates. Media were supplemented with antibiotics streptomycin (100 μg/mL), ampicillin (100 μg/mL) or chloramphenicol (30 μg/mL) when specified.

Serial plating of 1X PBS dilutions of feces in LB agar plates supplemented with the appropriate antibiotics were incubated overnight and YFP, CFP or mCherry-labeled bacterial numbers were assessed by counting the fluorescent colonies using a fluorescent stereoscope (SteREO Lumar, Carl Zeiss). The detection limit for bacterial plating was ~300 CFU/g of feces[9].

### In vivo evolution and competition experiments

All mice (*Mus musculus*) used in this study were supplied by the Rodent Facility at Instituto Gulbenkian de Ciência (IGC) and were given *ad libitum* access to food (Rat and Mouse No.3 Breeding (Special Diets Services) and water. Mice were kept at 20-24 °C and 40-60% humidity with a 12-h light-dark cycle. For the in vivo evolution experiment we used the gut colonization model previously established[9]. Briefly, mice drank water with streptomycin (5 g/L) only for 24 h before a 4 h starvation period of food and water. The animals were then inoculated by gavage with 100 μL of an *E. coli* bacterial suspension of ~10^8 colony-forming units (CFUs). Mice A2, B2, D2, E2, G2, H2 and I2 were successfully colonized with the invader *E. coli*, while mice C2 and F2 failed to be colonized. Six- to eight-week-old C57BL/6 J non-littermate female mice were kept in individually ventilated cages under specified pathogen-free (SPF) barrier conditions at the IGC animal facility. Fecal pellets were collected during more than one year (>400 days) and stored in 15% glycerol at −80 °C for later analysis. In the competition experiments between the invader ancestral *E. coli* and evolved populations, we colonized the mice using a 1:1 ratio of each genotype, with bacterial loads being assessed and frozen on a daily basis after gavage.

In vivo competition experiments in which the two modes of selection (directional and diversifying) were acting for a longer time period were performed using evolved invader *E. coli* populations colonizing mice D2, B2 and A2, H2. Here we used both male (n = 8) and female (n = 8) C57BL/6 J mice aged six- to eight-week-old treated with streptomycin during 3 days before gavage. *E. coli* populations evolving for short time periods do not allow for strong conclusions on which mode of selection is taking place. Evolved invader populations such as I2 or G2 were therefore not used for in vivo fitness assays. To assess the impact of the mouse resident *E. coli* in the competitive fitness of *dgoR* we performed one-to-one competitions between the invader ancestral and *dgoR* KO clones. We first homogenized the mice microbiotas by co-housing the animals during seven days. The animals (n = 6, female C57BL/6 J mice aged six- to eight-week-old) were then maintained under co-housing and given streptomycin-supplemented (5 g/L) water during seven days to break colonization resistance and eradicate their resident *E. coli*. At this point, the co-housed mice were removed from the antibiotic-supplemented water for two days. The following day, one group of mice was gavaged with an mCherry-expressing resident *E. coli* (n = 3 mice) while the other group (n = 3) was not, with all animals being individually caged from this point on and receiving normal water without antibiotic. The day after gavage, all mice were colonized with a mix (1:1) of the invader ancestral and the *dgoR* KO clones, and the bacterial loads were assessed and frozen on a daily basis.

### Microbiota analysis

Fecal DNA was extracted with a QIAamp DNA Stool MiniKit (Qiagen), according to the manufacturer's instructions and with an additional step of mechanical disruption[32]. 16 S rRNA gene amplification and sequencing was carried out at the Gene Expression Unit from Instituto Gulbenkian de Ciência, following the service protocol. For each sample, the V4 region of the 16 S rRNA gene was amplified in triplicate, using the primer pair F515/R806, under the following PCR cycling conditions: 94 °C for 3 min, 35 cycles of 94 °C for 60 s, 50 °C for 60 s, and 72 °C for 105 s, with an extension step of 72 °C for 10 min. Samples were then pair-end sequenced on an Illumina MiSeq Benchtop Sequencer, following Illumina recommendations. Sampling for microbiota analysis was performed until the microbiota composition stabilized (~1 year after the antibiotic perturbation).

QIIME2 version 2017.11[33] was used to analyze the 16 S rRNA sequences by following the authors' online tutorials (https://docs.qiime2.org/2017.11/tutorials/). Briefly, the demultiplexed sequences were filtered using the "denoise-single" command of DADA2 version 1.14[34], and forward and reverse sequences were trimmed in the position in which the 25th percentile's quality score got below 20. Diversity analysis was performed following the QIIME2 tutorial[35]. Beta diversity distances were calculated through Unweighted Unifrac[36]. For

taxonomic analysis, OTU were picked by assigning operational taxonomic units at 97% similarity against the Greengenes database version 13 (Greengenes 13_8 99% OTUs (250 bp, V4 region 515 F/806 R))[37].

## Whole-genome sequencing and analysis pipeline

DNA was extracted[38] from *E. coli* populations (mixture of >1000 clones) or a single clone growing in LB plates supplemented with antibiotic to avoid contamination. DNA concentration and purity were quantified using Qubit and NanoDrop, respectively. The DNA library construction and sequencing were carried out by the IGC genomics facility using the Illumina Miseq platform. Processing of raw reads and variants analysis was based on the previous work[39]. Briefly, sequencing adapters were removed using fastp version 0.20.0[40] and raw reads were trimmed bidirectionally by 4 bp window sizes across which an average base quality of 20 was required to be retained. Further retention of reads required a minimum length of 100 bps per read containing at least 50% base pairs with phred scores at or above 20. BBsplit (part of BBMap version 38.9)[41] was used to remove likely contaminating reads as explained previously[39]. Separate reference genomes were used for the alignment of invader (K-12 (substrain MG1655; Accession Number: NC_000913.2)) and resident (Accession Number: SAMN15163749) *E. coli* genomes. Alignments were performed via three alignment approaches: BWA-sampe version 0.7.17[42], MOSAIK version 2.7[43], and Breseq version 0.35.1[44,45]. Final average alignment depths for invader and resident populations across time points equalled 302 (median = 236) and 253 (median = 235), respectively. While Breseq provides variant analysis in addition to alignment, other variant calling approaches were used to identify putative variation in the sequenced genomes, and to verify data from Breseq. A naïve pipeline[39] using the mpileup utility within SAMtools version 1.9[46] and a custom script written in python was employed. Only reads with a minimum mapping quality of 20 were considered for analysis, and variant calling was limited to bases with call qualities of at least 30. At these positions, a minimum of 5 quality reads had to support a putative variant on both strands (with strand bias, pos. strand / neg. strand, above 0.2 or below 5) for further consideration. Finally, mutations were retained if detected in more than one of the alignment approaches, and if they reached a minimum frequency of 5% at a minimum of one time point sampled. Further simple and complex small variants were considered from freebayes version 0.9.21[47] with similar thresholds, while insertion sequence movements and other mobile element activity was inferred via is mapper version 2[48] and panISa version 0.1.6[49], as well as Breseq, as previously described[39]. All putative variants were verified manually in IGV version 2.7[50,51]. Raw sequencing reads were deposited in the sequence read archive under bioproject PRJNA666769. Population dynamics of lineage-specific dynamics and the resulting Muller plots were inferred manually and are meant strictly as a means of presenting the data. In order to generate these plots, mutations were sorted by frequency (descending for each time point at which the population was sampled). The largest frequency mutations were considered major lineages within which minor frequency mutations occurred. Assuming that a mutation, which arises subsequent to a preexisting mutation (an already differentiated lineage) cannot exceed the frequency of that preexisting mutation at any point, and will fluctuate in frequency with the preexisting one, we assigned mutations to the lineages within each population. While this resolved the majority of high frequency and medium frequency mutations, low-frequency mutations within the Muller plots cannot be placed with high confidence, and are only included for completeness.

## Prophage induction rate

To calculate the maximum prophage induction rate we grew *E. coli* lysogenic clones, starting with the same initial OD600 values: -0.1 (Bioscreen C system, Oy Growth Curves Ab Ltd), with agitation at 37 °C in LB medium in the presence or absence of mitomycin C along time

(5 μg/mL)[9]. The OD600 values were normalized by dividing the ones in the presence of mitomycin C by the ones in the absence of mitomycin C (sampling interval: 30 min). The LN of this ratios along time originates a lysis curve, where the maximum slope corresponds to the maximal prophage induction rate for each clone analyzed. We tested evolved clones from mouse A2, H2 and G2 against the ancestral clone which only carries the Nef and the KingRac prophages. We also tested clones of the resident strain that had evolved in the presence of the invader for more than 400 days (these clones were sampled from mouse A2).

## *E. coli* growth rate, growth curves, cell aggregation, biofilm and motility capacity

To calculate the maximum bacterial growth rate, we grew *E. coli* lysogenic clones, starting with the same initial OD600 values: -0.1 (Bioscreen C system, Oy Growth Curves Ab Ltd), with agitation at 37 °C in LB medium along time using reading intervals of 30 min. The LN of the OD600 values along time originates a growth curve, where the maximum slope corresponds to the maximum bacterial growth rate for each clone analyzed.

To test for metabolic differences of the *psuK/fruA* mutation, growth curves of evolved lysogenic *E. coli* clones, bearing the Nef and KingRac prophages, with or without the *psuK/fruA* mutation were performed with the same initial OD600 value (-0.03) for each clone. The clones were grown in glucose (0.4%) minimal medium (MM9-SIGMA) with or without pseudouridine (80 μM) and absorbance values were obtained using the Bioscreen C apparatus during 12 h.

Frozen stocks of *E. coli* clones were used to seed tubes with 5 mL of liquid LB. These were incubated overnight at 37 °C under static conditions to assess the formation of cell flocks/clumps, observable to the naked eye, in order to evaluate the formation of cell aggregates. Biofilm was tested according a previously published protocol[52] and to evaluate the motility capacity we adapted the protocol from Croze and colleagues[53]. Briefly, overnight *E. coli* clonal cultures grown with agitation at 37 °C in 5 mL LB medium supplemented with streptomycin (100 ug/mL) were adjusted to the same absorbance and a 3uL volume was dropped on top of soft agar (0.25%). Plates were incubated at 37 °C and photos were taken at day 1, 2 and 5 post-inoculation to assess swarming motility phenotype.

## Number of *E. coli* generations during mouse gut colonization

To estimate the number of generations of *E. coli* in the mouse gut, we used a previously described protocol to measure the fluorescent intensity of a probe specific to *E. coli* 23 S rRNA (as a measure of ribosomal content) that correlates with the growth rate of the bacterial cells[54]. We measured the number of generations of the ancestral *E. coli* clone while colonizing the gut of 2 mice, treated during 24 h with streptomycin (5 g/L) before gavage, during 25 days.

## Plasmid DNA extraction and PCR detection of ~69Kb (*repA*) and ~109Kb (*repB*) plasmids

Plasmid DNA was extracted from overnight cultures using a Plasmid Mini Kit (Qiagen), according to the manufacturer's guidelines. Specific primers for the amplification of *repA* and *repB* genes, were used to determine the frequency of the 68935 bp (-69 Kb) and 108557 bp (-109 Kb) plasmids, respectively, in the invader *E. coli* population.

The primers used for *repA* gene were:

*repA*-Forward: 5′-CAGTCCCCTAAAGAATCGCCCC-3′ and *repA*-Reverse: 5′-TGACCAGGAGCGGCACAATCGC-3′.

For *repB* the primer sequences were:

*repB*-Forward: 5′-GTGGATAAGTCGTCCGGTGAGC-3′ and *repB*-Reverse: 5′-GTTCAAACAGGCGGGGATCGGC3′.

PCR amplification of plasmid-specific genes was performed in 12 isolated random clones from mouse A2 at days 104 and 493. PCR

reactions were performed in a total volume of 25 μL, containing 1 μL of plasmid DNA, 1X Taq polymerase buffer, 200 μM dNTPs, 0.2 μM of each primer and 1.25 U Taq polymerase. PCR reaction conditions: 95 °C for 3 min, followed by 35 cycles of 95 °C for 30 s, 65 °C for 30 s and 72 °C for 30 s, finalizing with 5 min at 72 °C. DNA was visualized on a 2% agarose gel stained with GelRed and run at 160 V for 60 min.

## Construction of the *dgoR* KO mutant

P1 transduction was used to construct a Δ*dgoR* mutant (*dgoR* KO). This KO strain was created by replacing the wild-type *dgoR* in the invader ancestral YFP-expressing genetic background by the respective knockout from the KEIO collection, strain JW5627[55], in which the *dgoR* sequence is replaced by a kanamycin resistance cassette. The presence of the cassette was confirmed by PCR using primers dgoK-F: GCGATGTAGCGAGCTGTC, and yidX-R: GGGAATAAACCGGCAGCC. PCR reactions were performed in a total volume of 25 μL, containing 1 μL of DNA, 1X Taq polymerase buffer, 200 μM dNTPs, 0.2 μM of each primer and 1.25 U Taq polymerase. PCR reaction conditions: 95 °C for 3 min, followed by 35 cycles of 95 °C for 30 s, 65 °C for 30 s and 72 °C for 30 s, finalizing with 5 min at 72 °C. DNA was visualized in a 2% agarose gel stained with GelRed and run at 160 V for 60 min.

## RNA extraction, DNAse treatment, RT-PCR and qPCR

The Qiagen RNeasy Mini Kit was used for RNA extraction. RNA concentration and quality were evaluated in the Nanodrop 2000 and by gel-electrophoresis. DNase treatment was performed with the RQ1 DNase (Promega) by adding 0.5 μl of DNase to 1 μg of RNA and 1 μl buffer in a final volume of 15 ul, followed by incubation 30 min at 37 °C. Afterwards, 1 ul of stop solution was added and incubation for 15 min at 65 °C was performed to inactivate the DNase. As a control for complete DNA digest a PCR was performed on the reactions including positive controls. Reverse transcription was performed with M-MLV RT[-H] (Promega) by mixing 1 μg of RNA with 0.5 μl random primers (Promega) and nuclease free water to a volume of 15 μl, incubation at 70 °C for 5 min and a quick cool down on ice. Afterwards the reverse transcription was accomplished by adding 5 μl of RT buffer, 0.5 μl RT enzyme and 2 μl dNTP mix, followed by incubation for 10 min at 25 °C, 50 min at 50 °C and 10 min at 70 °C. The resulting cDNA was diluted 100-fold in nuclease free water before changes in gene expression were detected using the The QuantStudio 7Flex (Applied Biosystems) with iTaq Universal SYBR Green Supermix (BioRad) and the following cycling protocol: Hold stage: 2 min at 50 °C, 10 min at 95 °C. PCR stage (40 cycles): 15 s at 95 °C, 30 s at 58 °C, 30 s at 60 °C. Melt curve stage: 15 s at 95 °C, 1 min at 50 °C then increments of 0.05 °C/s until 95 °C. Melt curve analysis was performed to verify product homogeneity. All reactions included six biological and three technical replicates for each sample. A relative quantification method of analysis with normalization against the endogenous control *rrsA* and employing the primer specific efficiencies was used according to the Pfaffl method (add reference). The primers used were designed with PrimerQuest (idt). The used primer sequences were: *psuK* - TGCGTTAGCAGCGATTGA, A ATTTACGCCTGGTGGAGTAG; *arcA* - GATTCATGGTACGGGACAGTAG, CCGTGACAACGAAGTCGATAA; *yjtD* - CGCACATGGATCTGGTGATA, G GCGTGGCGTAGTAATGATA and *rrsR* - GTCAGCTCGTGTTGTGAAATG, CCCACCTTCCTCCAGTTTATC.

## Statistics and reproducibility

Correlation between microbiota diversity measures and *E. coli* loads (CFU) or persistence (1-presence or 0-absence) was performed in R using the statistical package rmcorr (version 0.5.2)[56] and lme4 (version 1.1-10)[57], respectively. The rate of accumulation of new ISs in vivo was compared using Wilcoxon paired signed ranked test for expected and observed insertions, while the rate of selective sweeps correlation was performed using the Spearman Correlation test. Selective sweeps were taken to be mutations or

HGT events that reached > 95% frequency in the population and kept high frequency until the end of the colonization. Statistical analysis of prophage induction as well as biofilm levels was performed using the Mann-Whitney test in GraphPad Prism (version 8.4.3). A single sample T-Test was used test if the growth rate of evolved invader clones deviates from the mean of the ancestral. A Wilcoxon rank sum test with continuity correction was used to compare the relative expression levels of the evolved clones with the ancestral. P values of < 0.05 were considered significant.

Pearson correlation tests between the frequency and the change in frequency of a mutation were performed to search for evidence of negative frequency-dependent selection. These were conducted for every mutation that showed parallelism and for each mouse, provided that the mutation was detected in at least four time points. The correlations were calculated in R with cor.test, used for the association between paired samples.

Linear mixed models (R package nlme, v3.1[58]) were used to analyze the temporal dynamics of the *dgoR* KO mutant frequency in the presence or absence of the resident *E. coli*. The frequency of the *dgoR* KO mutant was log10 transformed to meet the assumptions of parametric statistics.

Sample size in animal experiments was chosen according to institutional directives and in accordance with the guiding principles underpinning humane use of animals in scientific research. No data were excluded from the analysis. The experiments were randomized with animals being assigned arbitrarily to each experiment. The investigators were not blind towards the animal experiments.

## Statistics of time-resolved mutation frequency trajectories

The following statistics are designed to infer the prevalent type of selection from time-resolved mutant frequency data. Specifically, we use such data to discriminate adaptive evolution under directional selection, which can take place by periodic sweeps or by clonal interference, and adaptation under diversifying selection (to be defined below).

We use two test statistics for frequency trajectories of established mutants (*i.e.*, mutants that have overcome genetic drift):

- the frequency propagator $G(x)$, defined as the probability that a trajectory reaches frequency $x$,
- the sojourn time $T(x)$, defined as the time between origination at a threshold frequency $x_0$ and the first occurrence at frequency $x$, averaged over all trajectories reaching frequency $x$. In terms of the underlying coalescent, $T(x)$ is the time to the last common ancestor for a genetic clade of frequency $x$.

These observables discriminate the following modes of adaptive evolution:

- **Periodic selective sweeps under uniform directional selection.** This mode is characteristic of simple adaptive processes in small populations or populations with low mutational inputs, where adaptive mutations are rare enough to fix independently[59–61]. Almost all established adaptive mutations reach fixation, and sojourn times to an intermediate frequency $x > x_0$ are of order of their inverse selection coefficient (up to logarithmic corrections):

$$G(x) \approx 1, T(x) \sim \frac{1}{s}. \tag{1}$$

- **Clonal interference under uniform directional selection.** This mode occurs in asexual populations when adaptive mutations become frequent enough to interfere with one another[59–61]. Only a fraction of the established adaptive mutations reaches fixation; sojourn times to intermediate frequencies are set by a global

coalescence rate $\tilde{\sigma}$ that is higher than the typical selection coefficient of individual mutations[62]:

$$G(x) < 1, T(x) \sim \frac{1}{\tilde{\sigma}}. \tag{2}$$

Details of these dynamics depend on the spectrum of selection coefficients and on the overall mutation rate, which set the strength of clonal interference. For moderate interference, where a few concurrent beneficial mutations compete for fixation, we expect a roughly exponential drop of the frequency propagator, $G(x) \sim \exp(-\lambda x)$, reflecting the probability that a trajectory reaches frequency $x$ without interference by a stronger competing clade. Moderate interference generates an effective neutrality for weaker beneficial mutations and at higher frequencies[63]. This regime has been mapped for influenza[64]. In the asymptotic regime of a travelling fitness wave, where many beneficial mutations are simultaneously present, the fate of a mutation is settled in the range of small frequencies; that is, at the tip of the wave[65]. In this regime, emergent neutrality affects the vast majority of beneficial mutations and most of the frequency regime[66]. Hence, the frequency propagator rapidly drops to its asymptotic value $G(x=1) \ll 1$.

- **Adaptation under diversifying selection**. More complex selection scenarios involve selection within and between ecotypes, *i.e.*, subpopulations occupying distinct ecological niches[67,68]. An important factor generating niches and ecotypes is the differential use of food and other environmental resources. In this mode, ecotype-specific, conditionally beneficial mutations reach intermediate frequencies after a time given by their within-ecotype selection coefficients, but fixation can be slowed down or suppressed by diversifying (negative frequency-dependent) cross-ecotype selection[18],

$$G(x) \approx 1, T(x) \sim \frac{1}{s} \left( x \lesssim \frac{1}{2} \right) \tag{3}$$

$$G(x) < 1, T(x) \gg \frac{1}{s} (x \to 1). \tag{4}$$

The details depend on the details of the eco-evolutionary model (synergistic vs. antagonistic interactions, carrying capacities, amount of resource competition vs. explicitly frequency-dependent selection). In a model with directional selection within ecotypes, conditionally beneficial mutations rapidly fix within ecotypes, but lead only to finite shifts of the ecotype frequencies. In the simplest case, the resulting dynamics of ecotype frequencies is diffusive, resulting in an effectively neutral turnover of ecotypes[18]. Given negative frequency-dependent selection between ecotypes, fixations become even rarer and can be completely suppressed; that is, ecotypes can become stable on the time scales of observation. The separation of time and selection scales between intra- and cross-ecotype frequency changes is expected to be a robust feature of ecotype-dependent selection: sojourn of adaptive alleles to intermediate frequencies is fast, fixation is slower and rarer. In other words, ecotype-dependent selection is characterized by two regimes of coalescence times $T(x)$.

Frequency propagators and the coalescence time spectra expected under these evolutionary modes are qualitatively sketched in Supplementary Fig. 11. For periodic sweeps under directional selection (dark green, left column), $G(x)$ depends weakly on $x$ and $T(x)$ is set by rapid sweeps for all $x$. For clonal interference under directional selection (green, center column), $G(x)$ decreases substantially with increasing $x$ and $T(x)$ becomes uniformly shorter. Under negative frequency-dependent selection (brown, right column), $G(x)$ decreases substantially with increasing $x$, while $T(x)$ substantially increases for large $x$ and diverges in case of strong frequency-dependent selection generating stable ecotypes (dashed lines). (see Supplementary Fig. 11 for the results of simulations assuming a model of direction selection or assuming a resource competition model where ecotype formation occurs[31].

## The $p$-$\tau$ selection test

This test is based on qualitative characteristics of the functions $G(x)$, $T(x)$ and does not depend on details of the evolutionary process. We evaluate $G(x)$ and $T(x)$ for host-specific families of frequency trajectories; sojourn times are counted from an initial frequency $x_0 = 0.01$. Origination times at this frequency are inferred by backward extrapolation of the first observed trajectory segment; the reported results are robust under variations of the threshold $x_0$ and the extrapolation procedure. We then compute two summary statistics: the probability $p$ that a mutation established at an intermediate frequency $x_m$ reaches near-fixation at a frequency $x_f$,

$$p = \frac{G(x_f)}{G(x_m)}, \tag{5}$$

and the corresponding fraction of sojourn times,

$$\tau = \frac{T(x_f)}{T(x_m)}. \tag{6}$$

Here we use $x_m = 0.3$ and $x_f = 0.95$ to limit the uncertainties of empirical trajectories at low and high frequency; however, the selection test is robust under variation of these frequencies. We find evidence for different modes of evolution:

- The long-term frequency trajectories of mice B2, D2 and E2 are consistent with predominantly frequency-dependent selection (Fig. 2, Fig. 4a–c). The propagator $G(x)$ is a strongly decreasing function of $x$, resulting in fixation probabilities $p < 0.5$. Sojourn times $T(x)$ show two regimes with a stronger increase in the frequency range $x > 0.6$, as measured by time ratios $\tau > 3$.
- The trajectories of mice A2, G2, and I2 show a signature of recurrent selective sweeps and clonal interference under uniform directional selection (Fig. 4a–c). The propagator $G(x)$ is a decreasing function of $x$, resulting in fixation probabilities $p = 0.2 - 0.8$, depending on the strength of clonal interference. Fixation times are short, giving time ratios $\tau \lesssim 2$.
- The shorter trajectory of mouse H2 signals periodic sweeps under uniform directional selection (Fig. 3, Fig. 4a–c). The origination rate of mutations is lower than in the longer trajectories, and $G(x)$ shows a weak decrease with $p = 1$. Sojourn times $T(x)$ are short and grow uniformly with $x$, resulting in a time ratio $\tau = 2.25$. This pattern is expected under directional selection in the low mutation regime: $T(x) = \log[x/(1-x)]/s$ for individual mutations with a uniform selection coefficient $s$, leading to $\tau = 2.0$ for $x_m = 0.3$ and $x_f = 0.95$ (this value is marked as a dashed line in Fig. 4c).
- The trajectories of non-mutator lines in the long-term in vitro evolution experiment of Good et al[1], evaluated over the first 7500 generations, show an overall signal of clonal interference under uniform directional selection (Fig. 4c, Supplementary Fig. 12). The frequency propagators $G(x)$ are strongly decreasing functions of $x$ and sojourn times $T(x)$ grow uniformly with $x$. We find $p = 0.2 - 0.8$ and $\tau \lesssim 2$, similar to the pattern in mice A2, G2, and I2.

## Code for Selection tests

The code for selection tests from the mutation frequency trajectories can be found in the Supplementary Information file.

## Reporting summary

Further information on research design is available in the Nature Research Reporting Summary linked to this article.

## Data availability

The raw population sequencing and 16 s rDNA data generated in this study have been deposited in the open access sequence read archive (part of the National Center for Biotechnology Information) under accession code (Bioproject) PRJNA666769. Reference genomes were used for the alignment of invader (K-12 (substrain MG1655; Accession Number: NC_000913.2)) and resident (Accession Number: SAMN15163749) *E. coli* genomes. The Greengenes database (http://greengenes.lbl.gov) was used for microbiota taxonomic analysis.

## Code availability

The code for the Frequency-time statistics of mutation trajectories is available in the Supplementary Information file. The code for the simulations is available at: https://github.com/AmiconeM/2EvolutionModes and is linked to Zenodo database: https://doi.org/10.5281/zenodo.7043896 (https://doi.org/10.5281/zenodo.7043896).

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

## Acknowledgements

We thank Jonathan Howard for discussions, Arjan deVisser for a critical reading of the manuscript, and Benjamin Good for sharing data from in vitro experiments. We would also like to thank the personnel of the IGC's Rodent Facility, Genomic Facility and the Bioinformatics Unit for their assistance as well as Roberto Balbontin for the marked resident clone. N.F. was supported by "Fundação para a Ciência e Tecnologia" (FCT), fellowship SFRH/BPD/11075/2015, A.K. by a cooperation agreement between IGC and the University of Cologne. This work was also supported by project Global Gut Health Nature Research/Yakult Grant 623877 and ONEIDA project (LISBOA-01-0145-FEDER-016417) co-funded by FEEI – "Fundos Europeus Estruturais e de Investimento" from "Programa Operacional Regional Lisboa 2020", by national funds from FCT and FCT Project PTDC/BIA-EVL/7546/2020, and by Deutsche Forschungsgemeinschaft grant CRC 1310 (to M.L.).

## Author contributions

I.G. and N.F. designed and coordinated the study. N.F. performed the experiments with help from D.G. and E.S. A.K. performed the bioinformatic analysis. M.A. performed the simulations. M.L. developed the selection tests. I.G., N.F. and M.L. analysed the results and wrote the manuscript. All authors gave final approval for publication.

## Competing interests

The authors have declared no competing interests.
