## [Peer Review File · Nature Communications]

REVIEWERS' COMMENTS

Reviewer #1 (Remarks to the Author):

The authors have done a good job of adding more context on the novelty of the current study. I have no further comments.

Reviewer #2 (Remarks to the Author):

The manuscript modifications made by the authors clarify the key scientific advances of the study and extend our understanding of their results. The authors now clearly outline the novelty of their findings in the introduction and results, within the context of previously published results by their group. Specifically, the clock-like rate of adaptive evolution in *E. coli* throughout long-term evolution in a complex mammalian gut microbiome, the dynamics of phage domestication, and the emergence of distinct modes of evolution are now highlighted as the focal takeaways.

The authors now describe a plausible mechanistic explanation of how the presence of a related resident strain shifts the mode of evolution from diversifying to directional selection under the framework of metabolic niche overlap and phage transfer. The new Extended Data Figure 12 and accompanying statements in the results sufficiently describe their metabolic niche argument, primarily through the case study of *dgoR* (galactonate) knockout mutation in invader *E. coli*, which is only present when the resident *E. coli* is absent. This framework is also extended to potentially explain the parallel evolution of *frlR* (fructolysine metabolism regulatory gene) mutation across mice replicates due to the lack of overlap in metabolic capacity. Moreover, the authors now describe how the selective pressure to reduce prophage induction cost could lead to directional selection in the invading strain. The authors also now provide a brief description of conditions that would potentially favor clonal interference or periodic sweeps when directional selection is the dominant mode of evolution, as modelled in the new Extended Data Figure 10. Although this is not extensively rationalized (what causes the variations in mutational input of the invading strain or selective pressures across mice?), I believe that this is outside the scope of this paper, is grounds for future work, and should not be reason for further revision. Taken together, these additions strengthen the impact of their results.

Although there is a lack of addressing the effect of the background community on the evolution of the invading species, i.e. whether the presence of a related resident *E. coli* strain is sufficient to explain shifts in mode of adaptation, the authors rebuttal regarding infeasibility of a germ-free mice experiment for this manuscript due to length of time-scale is fair. Finally, minor technical details in methodology have now been addressed, with respect to exclusion of mouse I2 in *in vivo* competitive fitness assays, details of whole-genome sequencing, determination of lineage dynamics as visualized in Mueller plots, isolation of invading *E. coli* strains, and categorization of invading population by p - T statistics criteria.

The manuscript modifications and additional analyses sufficiently addressed my concerns.

Reviewer #3 (Remarks to the Author):

I thank the authors for their thorough responses to the review comments. In my opinion, the authors have satisfactorily addressed the comments.

RESPONSE TO REVIEWERS

REVIEWERS' COMMENTS

Reviewer #1 (Remarks to the Author):

The authors have done a good job of adding more context on the novelty of the current study. I have no further comments.

We thank the Reviewer for the comment.

Reviewer #2 (Remarks to the Author):

The manuscript modifications made by the authors clarify the key scientific advances of the study and extend our understanding of their results. The authors now clearly outline the novelty of their findings in the introduction and results, within the context of previously published results by their group. Specifically, the clock-like rate of adaptive evolution in *E. coli* throughout long-term evolution in a complex mammalian gut microbiome, the dynamics of phage domestication, and the emergence of distinct modes of evolution are now highlighted as the focal takeaways.

The authors now describe a plausible mechanistic explanation of how the presence of a related resident strain shifts the mode of evolution from diversifying to directional selection under the framework of metabolic niche overlap and phage transfer. The new Extended Data Figure 12 and accompanying statements in the results sufficiently describe their metabolic niche argument, primarily through the case study of *dgoR* (galactonate) knockout mutation in invader *E. coli*, which is only present when the resident *E. coli* is absent. This framework is also extended to potentially explain the parallel evolution of *frtR* (fructolysine metabolism regulatory gene) mutation across mice replicates due to the lack of overlap in metabolic capacity. Moreover, the authors now describe how the selective pressure to reduce prophage induction cost could lead to directional selection in the invading strain. The authors also now provide a brief description of conditions that would potentially favor clonal interference or periodic sweeps when directional selection is the dominant mode of evolution, as modelled in the new Extended Data Figure 10. Although this is not extensively rationalized (what causes the variations in mutational input of the invading strain or selective pressures across mice?), I believe that this is outside the scope of this paper, is grounds for future work, and should not be reason for further revision. Taken together, these additions strengthen the impact of their results.

Although there is a lack of addressing the effect of the background community on the evolution of the invading species, i.e. whether the presence of a related resident *E. coli* strain is sufficient to explain shifts in mode of adaptation, the authors rebuttal

regarding infeasibility of a germ-free mice experiment for this manuscript due to length of time-scale is fair. Finally, minor technical details in methodology have now been addressed, with respect to exclusion of mouse I2 in in vivo competitive fitness assays, details of whole-genome sequencing, determination of lineage dynamics as visualized in Mueller plots, isolation of invading E. coli strains, and categorization of invading population by p- τ statistics criteria.

The manuscript modifications and additional analyses sufficiently addressed my concerns.

We thank the Reviewer for the comments.

Reviewer #3 (Remarks to the Author):

I thank the authors for their thorough responses to the review comments. In my opinion, the authors have satisfactorily addressed the comments.

We thank the Reviewer for the comment.